When, why and how foot orthoses (FOs) should be prescribed for children with flexible pes planus: a Delphi survey of podiatrists

Dars Sindhrani darsy009@mymail.unisa.edu.au
Uden Hayley
Kumar Saravana
Banwell Helen A.
School of Health Sciences, Sansom Institute for Health Research, University of South Australia , Australia
Carpes Felipe
Electronic publication date: 2018 Apr 16
Publication date: 2018
Volume: 6
Electronic Location ID: e4667
Received 2018 Feb 27; Accepted 2018 Apr 4
Copyright: ©2018 Dars et al.
Copyright year: 2018
Copyright holder: Dars et al.
License: This is an open access article distributed under the terms of the Creative Commons Attribution License, which permits unrestricted use, distribution, reproduction and adaptation in any medium and for any purpose provided that it is properly attributed. For attribution, the original author(s), title, publication source (PeerJ) and either DOI or URL of the article must be cited.
License URL: https://creativecommons.org/licenses/by/4.0/

Keywords: Consensus, Delphi, Paediatric, Pes planus, Foot orthoses, Flat feet, And podiatrists.

Funding: The authors received no funding for this work.

==============================
Background

Flexible pes planus (flat feet) in children is a common reason parents and caregivers seek health professionals consult and a frequent reason podiatrists prescribe foot orthoses. Yet no universal agreement exists on the diagnosis of this condition, or when and how foot orthoses should be prescribed. The aim of this study was to garner consensus and agreement among podiatrists on the use of FOs for paediatric flexible pes planus.

Methods

A three round Delphi consensus survey was undertaken with 15 podiatry experts from Australia, New Zealand and the United Kingdom. Round One gathered consensus on the diagnosis and intervention into paediatric pes planus with specific questions on types of FOs and prescription variables used. Round Two and Three were based on answers from Round One and gathered agreement (rationale for choices) on a five point Likert scale. 70% of respondents had to agree to a statement for it to be accepted as consensus or agreement.

Results

Consensus and agreement was achieved for 83 statements directing the diagnosis of pes planus (using FPI-6 and/or rearfoot measures), common signs and symptoms (e.g., pain, fatigue, abnormal gait and other functional concerns) that direct when to intervene into paediatric flexible pes planus. Prefabricated orthoses were the preferred intervention where adequate control is gained with their use. When customised orthoses are prescribed, a vertical [heel] cast pour (71.4%) and minimal arch fill (76.9%) are the prescription variables of choice, plus or minus additional variables (i.e., medial heel (Kirby) skive, the use of a University of California Biomechanical Laboratory device or a medial flange) dependent on level of disorder and plane of excessive motion.

Conclusions

This study identified consensus and agreement on a series of diagnosis methods and interventions for the paediatric flexible pes planus. A clinical protocol was developed from the resultant consensus statements which provides clinicians with a series of evidenced-informed statements to better guide them on when, how and why FOs are used specific to this population.

Background

Flexible pes planus (flat feet) describes feet with a lowered medial longitudinal arch on weight bearing, that resolves when non-weight bearing (Fabry, 2010; Harris et al., 2004; Roth et al., 2013). It is distinct from rigid pes planus, a congenital, rigid or spastic deformity of the foot, affecting less than 1% of the population and often requiring surgical management (Harris et al., 2004; Rome, Ashford & Evans, 2010; WHO, 2016). Prevalence of paediatric pes planus ranges from 48.5% to 77.9% in children aged 2–16 years (Chen et al., 2014; Evans & Rome, 2011; Halabchi et al., 2013), reducing to only 2–23% in the adult population (Dunn et al., 2004; Golightly et al., 2012; Kosashvili et al., 2008). The larger prevalence observed in children can be explained by many factors, including age appropriate ligament laxity, and advancing maturation of neuromuscular control (Lin et al., 2001; Pfeiffer et al., 2006; Stavlas et al., 2005; Uden, Scharfbillig & Causby, 2017), with most of the arch ‘flatness’ reducing over the first decade of life (El et al., 2006; Rome, Ashford & Evans, 2010; Stavlas et al., 2005). However, despite being an expected finding in typically developing children (Uden, Scharfbillig & Causby, 2017) flexible paediatric pes planus is one of the most frequently cited orthopaedic concerns that prompt parents and caregivers to seek health practitioner advice (Fabry, 2010; Rome, Ashford & Evans, 2010). Whilst parental concern may stem from aesthetics, there is evidence that some children with flexible pes planus present with symptoms associated with their foot posture (i.e., symptomatic flexible pes planus). Children with symptomatic flexible pes planus have been shown to have reduced walking velocity (p < 0.05) (Kothari et al., 2015; Lin et al., 2001), poorer performance in lower-limb involved physical tasks (Lin et al., 2001), and are significantly more likely to have hip, knee (odds ratio = 1.33, p < 0.01) and back pain (odds ratio = 1.22, p = 0.01) (Kothari et al., 2016). Quality of life scores assessed by the Oxford Ankle Foot Questionnaire for Children (OxAFQ_C) highlight that children with symptomatic flexible pes planus have significantly lower scores when compared to children with ‘normal’ arches, particularly in the physical domain (66.7% vs 91.7%, p < 0.05) (Kothari et al., 2015). Furthermore, flexible pes planus that persists or presents in adulthood is associated with increased rates of intermittent lower back pain, anterior knee pain, joint degeneration, instability and functional limitations and/or disability (Dunn et al., 2004; Golightly et al., 2012; Kosashvili et al., 2008). The challenge for health professionals is to determine when intervention for paediatric flexible pes planus may be warranted to ameliorate symptoms and reduce complications later in life (Dunn et al., 2004; Golightly et al., 2012; Kosashvili et al., 2008).

The most frequently cited intervention for paediatric pes planus is foot orthoses (FOs) (Halabchi et al., 2013; MacKenzie, Rome & Evans, 2012; Rome, Ashford & Evans, 2010; Wenger et al., 1989; Whitford & Esterman, 2007). Other common non-surgical treatments include footwear, activity modification, weight reduction, manipulations, serial castings, and stretch exercises of gastrocnemius-soleal complex (Halabchi et al., 2013; Harris et al., 2004; Rome, Ashford & Evans, 2010), yet little is understood on how or when different interventions are used.

Minimal guidance exists for clinicians on diagnosis and intervention pathways specific to symptomatic presentations of paediatric flexible pes planus. The paediatric flatfoot proforma (p-FFP), developed by Evans (2008), offers direction when management may be required (Evans, 2008), however, there is no reference standard on what symptoms of flexible pes planus should be cause for intervention, or indeed, no consensus on what intervention methods should be used. Without appropriate tools to guide practice, clinicians are guided by their clinical experience and judgement only, potentially resulting in disparity amongst professionals and outcomes of management alike. This identifies the need for developing a clinical protocol that not only identifies the commonly encountered signs and symptoms of paediatric flexible pes planus but also helps to identify the treatment options being utilised in the clinical practice.

In the absence of appropriate evidence-informed tools, consensus surveys such as the Delphi method, are a reliable and valid method of determining expert opinion (Van der Linde et al., 2005; Vernon, 2009). This study primarily aimed to garner consensus and agreement, with Delphi consensus survey, from experienced podiatrists of Australia (AU), New Zealand (NZ) and United Kingdom (UK), on the presentation and management of paediatric flexible pes planus. The secondary aim was to develop a clinical protocol based on this expert opinion to direct clinicians on when, why and how FOs should be prescribed for children with symptomatic flexible pes planus.

Methods

Study design

The study was a three-round modified Delphi panel design where participants’ opinion was sought in Round One, with responses collated and analysed for consensus (Hasson, Keeney & McKenna, 2000; Hsu & Sandford, 2007; Vernon, 2009). Responses not reaching consensus were reviewed by participants in successive rounds for consideration, commenting and ranking of agreement (Fig. 1). The study was approved by the University of South Australia’s Human Research Ethics Committee (Protocol no. 0000035501). All participants provided written informed consent.

Figure 1 Flow diagram of Delphi process with number of statements in each round.

Participants

Fifteen podiatrists were recruited for the Delphi survey panel. Potential participants were required to be a registered and practicing podiatrist with clinical experience of ≥10 years, or have worked in a paediatric-focused position for ≥8 years. The participants’ inclusion was also based on satisfying at least one of the following criteria: held an academic position teaching paediatric podiatry/clinical practice within a podiatry program; held a clinical position with practice focused on paediatric assessment and intervention; or had published research on paediatric theory/FOs within past five years.

Participants were recruited from Australia, New Zealand and the United Kingdom. These three countries were chosen due to similarities in podiatry undergraduate education, scope of podiatric practice and health care contexts. A total of 38 potential participants were identified based on the above criteria, international reputation, employment within ‘paediatric’ specific public health roles, involvement in published research on paediatric podiatry/FOs, and staff listings of academic institutes. Participants were then randomly selected from this list of potential participants and invited to participate until 15 experts were enrolled. This number of participants offers a broad sample of podiatry expertise whilst remaining pragmatic as a manageable panel size (Hasson, Keeney & McKenna, 2000; Hsu & Sandford, 2007; Van der Linde et al., 2005; Vernon, 2009).

Procedure

Potential participants were invited to participate (Data S1). A preliminary survey was completed to ensure eligibility for the study and informed consent to participate was obtained electronically. Once enrolled, participants were given four weeks to complete each round. Late responders were sent a reminder with further two weeks’ extension. Participants were considered non-responders if they failed to complete the survey within six weeks of the distribution date and had not requested extra time. The participants were reminded in each round that the focus of this study was on flexible pes planus in otherwise typically developing children, i.e., not associated with neurological, muscular or structural disease or abnormalities.

Survey format

A Delphi survey consisting of three rounds was conducted (Fig. 1). Data was collected using online survey platform Survey Monkey® (SurveyMonkey Inc., Palo Alto, CA, USA). Round One was developed based on common assessment questions including child’s subjective history, as described in the GALLOP assessment tool developed (Cranage, Banwell & Williams, 2016) and common prescription variables for foot orthoses as established from previous work in an adult population (Banwell et al., 2014). Prior to conducting the survey, the preliminary survey and Round One was piloted by two Australian podiatrists independent to study, for feedback on structure, clarity, ordering and framing of the questions. These podiatrists were excluded from the main study. All three Delphi rounds were divided into four sections: establishing the presence of flexible pes planus; intervention into flexible pes planus; using FOs for flexible pes planus in children; and approach to prescription of FOs for flexible pes planus in children (Data S2). Section one and two included questions on the method of clinical diagnosis and signs and symptoms associated with paediatric flexible pes planus that may initiate intervention. Section three and four included questions on participants’ preference for the FOs prescribed for this population, including when and how customised or pre-fabricated orthoses would be used and when alternative interventions may be considered (e.g., footwear modification and rearfoot wedges). Specific questions on their use of different prescription variables were also included, with participants asked to describe their thoughts on desired outcomes of FOs intervention for paediatric flexible pes planus. Participants were supplied a glossary of terms to accommodate variations in terminology between countries (Data S3).

Round One included a total of 31 questions requiring participants to supply closed and open-ended responses. The closed-ended questions were directly analysed for consensus by two authors (SD, HB). The open-ended responses were themed into statements and analysed for consensus by the same two authors (SD, HB), with recommendations from the third author (HU) if disagreement arose. Statements were considered to have reached consensus within Round One when 70% or more of participants indicated the same statement. Statements were required to receive 70% consensus to be accepted, thus remaining consistent with existing literature (Banwell et al., 2014; Bisson et al., 2010; Cranage, Banwell & Williams, 2016; Mokkink et al., 2010; Vernon, Parry & Potter, 2003). Round Two was based on the statements developed from Round One (Fig. 1). Participants were requested to consider each question in relationship to the presentation of a child between the ages 0–18 with flexible pes planus and indicate their level of agreement on a five point Likert scale (i.e., strongly disagree, disagree, neutral, agree or strongly agree) (Likert, 1932), and comment further if desired. As an example, statements related to the likelihood of prescribing FOs in the presence of pain are displayed in Fig. 2.

Figure 2 An example question from Round Two.

In Round Two and Three, statements were considered accepted if 70% or more of participants indicated that they agreed or strongly agreed with a statement. Statements not reaching 50% agreement were excluded (Cranage, Banwell & Williams, 2016; Okoli & Pawlowski, 2004). Statements receiving 50–69% agreement were reviewed in subsequent survey rounds, when available, to ensure adequate panel consideration. Statements were also excluded if agreement had not been achieved within two rounds.

The a priori decision was that the Delphi would be concluded when the response rate dropped below 70% or when Round Three was completed, irrespective of agreement. Participants were asked to keep their involvement confidential and participants were asked to maintain intra-panel communication anonymity throughout the survey.

Data management and analysis

The outcomes of interest were consensus and agreement. Consensus was sought in Round One only (Fig. 1). Consensus was achieved with ≥70% participants (≥11 of 15 participants) indicated a consistent response for the open and closed ended questions (Data S2). Agreement was sought in Rounds Two and Three (Fig. 1). Agreement was achieved when ≥70% (≥10 of 14 participants) agreed or strongly agreed to a given statement based on a five-point Likert scale (Fig. 1).

The further comments received in Round One and Two were collated and themed into statements by two authors (SD and HB) to be reviewed in subsequent rounds. Participants were provided with tabled presentations of the data from previous rounds, an example of this can be seen in Fig. 3.

Figure 3 An example question from Round Three.

Data not relating to the scope of the study, i.e., flexible pes planus in otherwise normally developing children and conservative management, were excluded. Statements not reaching consensus or agreement by the end of Round Three were also excluded (Data S4). Descriptive data analysis was undertaken in Microsoft Excel 2016 (Microsoft Corporation, Redmond WA, USA).

Results

Participant’s characteristics

Nine of the participants were Australian (64.3%), three were from the United Kingdom (21.5%) and two from New Zealand (14.5%). Overall, 57% of participants were male (8:6 males:females ratio) with an average age of 39.1 years (SD 4.8, range = 33–46 years). Participants had been practicing for an average of 16.9 years (SD 5.6, range 8–28 years) and 78.6% either held or were working towards a recognised post-graduate qualification. Over 70% of the panel (71.4%) listed more than one employment setting with clinicians (n = 11) and academics (n = 9) reaching almost equal representation. Four participants also identified themselves as researchers. Only two participants identified themselves as academics only (Table 1).

Table 1 Participants’ characteristics.

Gender	8 Males	57.1%	
6 Females	42.8%	
Mean practice duration (mean + range)	16.9 years	8–28 years	
Highest qualification	7 PhD	50%	
2 Master’s Degree	0.14%	
1 Graduate certificate	0.07%	
3 Bachelor’s Degree	0.21%	
Primary position	6 clinicians	42.8%	
6 academics	42.8%	
2 researchers	0.14%	
Secondary position	5 clinicians	0.36%	
3 academics	0.21%	
2 researchers	0.14%	
4 No secondary position	0.28%	
Estimated average paediatric patient load (mean + range)	57.9%	20–100%	
Estimated average paediatric consultations per week (mean + range)	16 children	2–50 children	
Estimated average number of orthosis prescribed per week for children/adolescents	85.7%	1–5 pairs	
0.07%	6–10 pairs	
0.07%	11–15 pairs	

Survey findings

Consensus

Round One received 100% response rate (15/15) and resulted in 21 statements reaching consensus and 173 statements to be reviewed in subsequent rounds (Table 2). Consensus was reached for the diagnosis of paediatric flexible pes planus to be determined using an assessment tool such as the Foot Posture Index—six item version (FPI-6) or the Paediatric Flat Foot Performa (pFFF) (80.0% consensus), or by visually assessing static foot posture (73.3% consensus) including resting and neutral calcaneal position (84.6% consensus). For determination of foot function in paediatric flexible pes planus population, three techniques reached consensus: range of motion (100% consensus); visual gait analysis (93.3% consensus); and muscle strength (93.3% consensus), (Table 2). The signs and symptoms that were considered to increase the likeliness of FOs prescription were foot pain (93.3% consensus), lower leg pain (73.3%), activity limitation (73.3% consensus) and a severe abnormal foot posture, such as two standard deviations from the expected measure (78.6% consensus). Conversely, the panel collectively agreed that that the age or weight of the child does not influence the decision to use FOs (71.4% agreement and 92.3% consensus respectively), (Table 2).

Table 2 Statements reaching consensus (>70%) in Round One.

Category	Statement	Level of consensus	
Determination of paediatric flexible pes planus;	Visual/measured static foot posture assessment	73.3%	
Foot posture tools (e.g., Foot posture index (FPI), Paediatric flat foot proforma (pFFF))	80.0%	
Static foot posture measures;	Rearfoot position (Resting Calcaneal Stance Position–RCSP & Neutral Calcaneal Stance Position–NCSP)	84.6%	
Foot function determination in paediatric flexible pes planus;	Visual gait analysis	93.3%	
Range of motion assessment	100%	
Muscle strength assessment	93.3%	
Likeliness of FOs prescription for paediatric flexible pes planus;	Severe abnormal foot posture (two Standard Deviations from expected measure)	78.6%	
Activity limitation	73.3%	
Foot Pain	93.3%	
Lower limb pain	73.3%	
Weight/mass of the child appropriate to initiate FOs treatment for flexible pes planus;	Weight does not influence the treatment decision	92.3%	
Prescription variables used for customised FOs for flexible pes planus;	Neutral/vertical cast pour	71.4%	
Minimal arch fill	76.9%	
Prescription variables NOT to be used (0% use) for customised FOs for flexible pes planus;	Blake inverted device (>15 degrees)	84.6%	
Everted cast pour	91.6%	
Blake inverted rearfoot post (>15 degrees)	90.0%	
Everted rearfoot post	90.0%	
Rearfoot post with motion	88.9%	
Maximum arch fill	72.7%	
Inverted forefoot post	70.0%	
Everted forefoot post	77.8%	

For management of paediatric flexible pes planus, the use of pre-fabricated FOs was preferred over customised FOs (74.0 ±7.1%, range 5–100 vs 34.0 ±9.3%, range 0–100), (Table 2). In the event that customised FOs are prescribed, the specific prescription variables that should be used are; a neutral or vertical (heel) cast pour (71.4% consensus), and a minimal arch fill (76.9% consensus), (Table 2). Whereas eight prescription variables received negative consensus, that is, they should not be used for customised FOs prescribed for paediatric flexible pes planus (Table 2). These variables were: maximum arch fill (72.7% consensus); everted cast pour (91.6% consensus); everted rearfoot post (90.0% consensus); Blake inverted device [>15°] (84.6% consensus); Blake inverted rearfoot post [>15°] (90.0% consensus); rearfoot post with motion (88.9% consensus); and inverted or everted forefoot post (70.0% and 77.8% consensus respectively). No further consensus was achieved.

Agreement

Round Two and Three both received 93.3% response rate (14/15), (Fig. 1). Of the 152 statements reviewed in Round Two, 44 achieved agreement (Table 3), 36 received between 50–69% agreement, 72 were excluded (Data S4) and seven new statements were generated from comments received (Fig. 1). Of the 43 statements reviewed in Round Three, 18 statements achieved agreement (Table 3), all other statements were excluded, (Fig. 1). No further comments were sought in Round Three (Fig. 1).

Table 3 Statements receiving agreement of  >70% from Round Two and Three of Delphi.

Category	Statement	Agreement	
Flexible pes planus determination;	Visual assessment of dynamic foot in gait	85.7%	
Dynamic WB and non-WB foot motion and/or measures	85.7%	
Static foot posture measures;	Foot Posture Index 6 (FPI-6)	100%	
Foot function determination in paediatric flexible pes planus;	Neurological assessments (Reflexes, sensation, tone and strength)	78.5%	
Single Limb Balance	71.4%	
The Balance tests to assess foot function;	Hopping (n = dominant and non-dominant leg)	78.5%	
Timed balance, standing on one leg (eyes open & closed)	85.7%	
All balance tests for comprehensive assessment of functional impact rather than pes planus presence	100%	
Walk along straight line/marching/heel-toe gait (forwards and backwards)	78.5%	
Running	78.5%	
Jumping	71.4%	
Likeliness of FOs prescription for paediatric flexible pes;	If dynamic foot function affected (instability in single leg stance, walking, running, turning, etc.)	85.7%	
In presence of symptoms (pain, reduced function, strength and structure per WHO-ICD)	100%	
In presence of structural changes (hallux abducto valgus, hallux limitus, etc.)	71.4%	
With foot posture related delayed milestones	78.5%	
With parental concern, accompanied by affected function	78.5%	
With gross pronation (apropulsive gait and low tone)	100%	
With hereditary lower limb disorder/s changing function and causing pain	92.8%	
If improvement in ICF (The International Classification of Functioning, Disability and Health) outcomes	71.4%	
Symptoms (e.g., pain, general discomfort, reduced walking, poor endurance and balance)	100%	
Plantar arch/fascia pain	92.8%	
Heel pain	78.5%	
Tibialis Posterior tendon pain	100%	
Medial Tibial Stress Syndrome (MTSS) type symptoms	100%	
Activity related pain	92.8%	
Regarding child’s age, decision of FOs use is influenced by:	Other factors than age as extent/degree of deformity, type and frequency of activity, and function	92.8%	
Acquisition of motor skills rather than age	71.4%	
FOs preferred, in:	Presence of symptoms (foot and leg pain, affected function and gross motor skill development)	92.8%	
The aim of prescribing FOs is to:	Reduce symptoms	92.8%	
Reduce fatigue	85.7%	
Improve gross motor skill	85.7%	
Improve balance, stability, comfort, coordination, stamina and endurance	92.8%	
Improve overall wellbeing and health outcomes per WHO-ICF, thus improved quality of life	71.4%	
When comparing pre-fabricated FOs to custom-made FOs;	Pre-fabricated FOs are easily modifiable	78.5%	
Pre-fabricated FOs are cost effective	71.4%	
Pre-fabricated FOs should be used when they offer enough control	71.4%	
Customised FOs should be used if pre-fabricated FOs do not provide adequate support for the child’s foot	100%	
Pre-fabricated FOs can be quickly dispensed i.e., as soon as the parents decide to use them	78.5%	
The features that guide the choice of prefabricated FOs specific may include:	Easy fit in a shoe	71.4%	
Smooth contours (low irritation and increased comfort)	71.4%	
Material easily customised	71.4%	
Appropriate material strength to provide needed control	85.7%	
Financial limitation of parents/cost	71.4%	
Size availability	78.5%	
For Custom FOs, a Medial (Kirby) heel skive may be used:	To provide additional/better rearfoot control	78.5%	
To help reduce STJ pronation	85.7%	
In severe pes planus in the frontal plane	71.4%	
For custom FOs, a UCBL (i.e., Medial and Lateral flange) device may be used:	In grossly pronated feet with hypotonia	71.4%	
When extra mid foot control is required in transverse plane	92.8%	
For custom FOs, a medial flange device may be used:	When extra midfoot control is required	92.8%	
To limit MTJ pronation and prevent foot rolling over device	78.5%	
In very flexible pes planus where medial edge of device is not tolerated	71.4%	
Shell materials for Custom FOs;	Three-dimensional printing materials	71.4%	
Alternative devices for flexible pes planus;	Rearfoot or heel wedges/lifts	71.4%	
Exercise therapy	85.7%	
For custom FOs, consider;	Adequate accommodation of talo-navicular region to prevent blistering by wider midfoot area in device	71.4%	
Notes.

WB Wight bearing

WHO-ICD World Health Organisation-International Classification of Diseases

STJ Subtalar joint

MTJ Midtarsal joint

UCBL University of California Biomechanics Laboratory

Of the 62 accepted statements, eight reached 100% agreement (Table 3). These were: use of Foot Posture Index–6 (FPI-6) to determine the presence of flexible pes planus; conducting balance tests to determine the functional impact of pes planus; and to prescribe FOs in presence of symptoms correlating to WHO-ICD (World Health Organisation International Classification of Diseases), gross pronation due to an apropulsive gait and low tone, and also other symptoms such as general discomfort, foot and leg pain, reduced walking ability, poor endurance and balance,  affected function and gross motor skills, pain of the Tibialis Posterior tendon and Medial Tibial Stress Syndrome (MTSS) type symptoms (Table 3).

There was high agreement that flexible pes planus may be assessed by visual assessment of foot in gait with dynamic weight bearing and non-weight bearing foot measures (85.7% agreement) (Table 3). The techniques and balance tests to assess foot function in pes planus included: neurological assessment (78.5% agreement); single limb balance (71.4% agreement); hopping (78.5% agreement); running (78.5% agreement); jumping (71.4% agreement); walking along a straight line with marching and heel-to-toe gait (78.5% agreement); and timed balance test with eyes opened and closed (85.7% agreement).

Other pain or symptoms that increase the likeliness of intervention included: activity related pain (92.8% agreement); affected dynamic foot function (85.7% agreement); presence of structural changes (71.4% agreement); delayed milestones (78.5% agreement); hereditary lower limb conditions causing pain or changing function (92.8% agreement); and pain in plantar fascia or heel (92.8% and 78.5% agreement respectively). Whereas parental concern would only result in prescription of FOs if accompanied by affected function (78.5% agreement), (Table 3).

The panel also agreed on prescribing FOs for paediatric pes planus based on the extent of pathology, degree of deformity, acquisition of motor skills, activity levels and function rather than a specific age of the child (92.8% agreement) (Table 3). The aim of prescribing FOs, that is expected outcomes of treatment that were agreed upon, included: reduction of symptoms and fatigue (92.8% and 85.7% agreement respectively); improving gross motor skills (85.7% agreement); improving balance, stability, comfort, coordination, stamina and endurance (92.8% agreement); and overall wellbeing and health outcomes which will in turn increase the overall quality of life for children with flexible pes planus (71.4% agreement). The rationale behind preferring pre-fabricated orthosis over customised were pre-fabricated FOs being easily modifiable (78.5% agreement), cost effective (71.4% agreement) and require less time to dispense (78.5% agreement). Other desirable features of pre-fabricated FOs identified include good fitting in shoes (71.4% agreement), smooth contour thus increased comfort (71.4% agreement), size availability (78.5% agreement), and appropriate strength of material to provide adequate control (85.7% agreement). Furthermore, participants were unanimous in agreeing (100%) that customised FOs should be used when prefabricated devices do not supply adequate support for the child’s foot (Table 3).

When customised FOs are to be used, a medial (Kirby) heel skive can be prescribed when severe pes planus exists in frontal plane (71.4% agreement), to provide better rearfoot control (78.5% agreement), or to reduce subtalar joint pronation (85.7% agreement) (Table 3). Agreement was reached that a UCBL device (which has both medial and lateral flange in-situ) can be used in the presence of grossly pronated feet with hypertonia (71.4% agreement), and to provide midfoot control for transverse plane motion (92.8% agreement). A medial flange alone could be prescribed to prevent the foot rolling over the device or if the aim of the device is to limit midtarsal joint pronation (78.5% agreement) or when extra midfoot control is required (92.8% agreement) or if the child cannot tolerate the medial edge of the device due to having very flexible pes planus (71.4% agreement). It was also agreed to accommodate talo-navicular bulge by including a wider midfoot area in the customised FOs (71.4% agreement). Other alternative devices used to treat flexible pes planus as agreed upon, include rearfoot or heel wedges (71.4% agreement) and exercise therapy (85.7% agreement). Over 70% of the panel also agreed on using three-dimensional printing shell materials for customised FOs when available (71.4% agreement), (Table 3).

At the completion of the three rounds, 83 statements (21 consensus and 62 agreement) were accepted on when, why and how orthoses are prescribed for paediatric flexible pes planus (Fig. 1, Tables 2 and 3).

Development of consensus-based clinical protocol

Following analysis of the Delphi survey findings, all included consensus and agreement statements were compiled to construct ‘A clinical protocol for paediatric flexible pes planus’ (Fig. 4). The protocol consists of three main sections: Confirm diagnosis; Signs and symptoms; and Intervention.

Figure 4 A protocol for paediatric flexible pes planus.

The protocol firstly allows clinicians to record the method of diagnosis (Fig. 4). Secondly, using a simple ‘tick and flick’ flow chart, the observed and reported signs and symptoms can be indicated. The symptom of pain is sub-divided into different regions, based on the agreement of the expert panel, with functional symptoms such as reported fatigue and perceived excessive tripping also included (Fig. 4). A section for identifying different signs specific to flexible pes planus and its history was also included for clinicians to tick appropriate options. These included: gross pronation; gait abnormalities; reduced range of motion (ROM); reduced muscle strength; activity limitation; affected function during single leg stance, walking, running and turning; poor endurance and balance; diagnosed Developmental Coordination Disorder (DCD); structural changes; delayed milestones and gross motor skill acquisition; neurological concerns including absent reflexes, affected sensation and low tone; and hereditary limb disorders changing foot function.

Finally, the protocol directs clinicians to consider prefabricated FOs in the first instance, confirming adequate fit and control is achieved, with recommendations for the prescription of customised FOs if this is not achieved (Fig. 4). Other alternative or additional conservative interventions or treatments are listed to direct the clinician to consider: footwear modification; activity modifications; strengthening exercises and stretching exercises where required.

This clinical protocol can be used concurrently with alternative tools such as the pFFP by Evans (2008) and Harris et al. (2004) flow chart, to help clinicians follow a logical pattern for management of paediatric flexible pes planus.

Discussion

Despite its common presentation in clinical practice, to date, there has been limited research which has systematically investigated how best to assess, diagnose and treat paediatric flexible pes planus. This has resulted in persistent knowledge gaps in this area. This research aims to address this knowledge gap by gathering expert international podiatry opinion on the presentation, diagnoses and intervention for paediatric flexible pes planus. The panel concluded that age and weight are not influencing factors for intervention, however, the presence of symptoms, degree of deformity or determinants of function were influential. When management is required, pre-fabricated orthoses are preferred by clinicians if they provide adequate support. Customised FOs, with targeted prescription options, should be prescribed when support is not adequately achieved with pre-fabricated devices.

While the literature acknowledges paediatric flexible pes planus as a frequently observed concern, there is considerable debate on its diagnosis (Pfeiffer et al., 2006; Roth et al., 2013; Tudor et al., 2009), and whether or not intervention is required (Pfeiffer et al., 2006; Tudor et al., 2009; Whitford & Esterman, 2007). A 2010 Cochrane systematic review on the efficacy of non-surgical intervention for paediatric flexible pes planus (Rome, Ashford & Evans, 2010) concluded that very limited high level evidence exists on this topic with equivocal findings for intervention, specifically FOs. However, this review was based on only three randomised control trials (Powell, Seid & Szer, 2005; Wenger et al., 1989; Whitford & Esterman, 2007), with only one of the studies using a symptomatic population. Indeed, much of the research on the use of FOs for flexible pes planus in children, to date, has been focussed on non-symptomatic populations (Aboutorabi et al., 2014; Bleck & Berzins, 1977; Bok et al., 2014; Bordelon & Lusskin, 1980; Capasso, 1993; Riccio et al., 2009; Valmassy & Terrafranca, 1986). The findings of this research demonstrate that it is the presenting symptoms of a child with flexible pes planus that guide the podiatrists’ decision to intervene along with clinical signs of the condition (observations and measurements). Furthermore, this research has determined that pain is only one of the symptoms that should initiate treatment, with presentations of: activity limitation; fatigue; perceived excessive tripping; and decline in function, strength and endurance also contributing to ‘symptomatic’ flexible pes planus. These findings are in line with previous literature (Evans, 2008; Halabchi et al., 2013; Harris et al., 2004; Whitford & Esterman, 2007), which found that children with flat feet, when compared to children with typically developing arches, have lower physical performance in tasks including squatting and standing, standing on toes, toe walking, heel walking and one leg standing and hoping (p < 0.02) (Lin et al., 2001) and experience increased knee (odds ratio = 1.33, p < 0.01), hip and back pain (odds ratio = 1.22, p = 0.01) (Kothari et al., 2016); which negatively affects their quality of life (Kothari et al., 2015). As such the aim of intervention, as agreed by panellists, was to reduce these frequently observed symptoms which potentially increases the overall quality of life for children with symptomatic flexible pes planus.

It was determined that interventions for pes planus was inclusive of FOs, footwear changes, activity modifications and stretching and strengthening exercises. Within the literature and anecdotally, FOs are often cited as a frontline management strategy for pes planus (Halabchi et al., 2013; MacKenzie, Rome & Evans, 2012; Rome, Ashford & Evans, 2010; Wenger et al., 1989; Whitford & Esterman, 2007). There is a lack of universally accepted pathway that defines when and how foot orthoses should be used in the management of paediatric flexible flatfeet (Stavlas et al., 2005; Tudor et al., 2009). Assessment tools, such as the p-FFP (Evans, 2008) and Harris et al. (2004) clinical pathway, assist in guiding the practitioner, however the p-FFP does not identify the specific intervention modalities and Harris et al. (2004) includes rigid pes planus and a surgical focus to their interventions. This protocol adds to the existing literature by detailing the specific signs and symptoms that are presented in the clinical practice and detailing specifics on intervention. For example, this research determined that pre-fabricated orthoses are preferred by participants due to cost effectiveness, timeliness of dispense and ease of modification. Customised FOs, however, were recommended when adequate support was not gained with the use of prefabricated devices, and with targeted prescription dependant on the ‘level of control’ required. Despite this preferential reasoning, it is importance to acknowledge that limited evidence exists for the use of one type of FOs over another (MacKenzie, Rome & Evans, 2012; Rome, Ashford & Evans, 2010), with no known studies comparing the impact of different prescription variables within a paediatric population. The results of this research may assist in developing future research which have a particular focus on informing clinical decision making. Specifically, on which prescription variables are frequently used by clinicians for symptomatic pes planus. Moreover, selecting specific prescription variables depending on different features of the deformity. As a starting point for translating best available evidence into clinical practice, this research has developed a clinical protocol which can offer clinicians, researchers and other stakeholders’ in this field opportunities for evidence-informed assessment, diagnosis and intervention for paediatric flexible pes planus.

There are some limitations to this Delphi survey. Firstly, the absence of agreement on a universally accepted definition of paediatric flexible pes planus. It was clearly established at the commencement of the study that the aim was a non-pathological flexible pes planus in children that was not associated with any muscular, neurological or osseous abnormalities. Secondly, even though this research provides evidence of consistency in podiatric practice for paediatric flexible pes planus, being a Delphi survey methodology, it is considered as an expert opinion only. As the participants were selected from three different countries with extensive clinical expertise, this research sought to capture broad and diverse opinions. Moreover, participants were not required to declare any potential conflicts of interest (including vested interests in foot orthoses manufacture/brand ownership), however, the research questions were framed around specific features of FOs only, excluding specific ‘brands’. This is specifically prudent given two sets of participants reported sharing a common place of employment. As part of the research process, participants were reminded of the importance of academic rigor in terms of remaining anonymous, strategies were put in place to minimise collusion and the definition of “expert” was based on current podiatry literature (Banwell et al., 2014; Okoli & Pawlowski, 2004; Vernon, 2009). Furthermore, due to limited psychometric testing of individual diagnostic measures used for paediatric pes planus (Evans, Nicholson & Zakarias, 2009), the measures indicated in this research should be used with caution and with clinical experience and judgement.

Conclusion

Flexible paediatric pes planus presents a dichotomy in clinical practice. While it is commonly encountered, to date there has been minimal guidance on how best to assess, diagnose and treat it. This research, by bringing together experienced podiatrists in this field from across three different counties, has generated expert-informed statements which can be used to guide clinical practice. In order to facilitate timely and effective translation to clinical practice and promote evidence-based practice, a ready-to-use clinical protocol provides clinicians with the opportunity to complement their clinical expertise with current best available evidence while sharing the decision making with parents and caregivers.

Supplemental Information

Data S1 Information sheet and invitation email sent to participants

Click here for additional data file.

Data S2 Round 1 of Delphi survey

Click here for additional data file.

Data S3 Data S4—Glossary of terms (Prescription variables)

Click here for additional data file.

Data S4 All excluded statements

Click here for additional data file.

Data S5 Raw data from all rounds of delphi survey

Click here for additional data file.

The Authors would like to thank the podiatrists involved with piloting the surveys and sincerely acknowledge the contribution of the expert panel towards this research.

Abbreviations

FOs Foot Orthoses

OxAFQ_C Oxford Ankle Foot Questionnaire for Children

pFFP The paediatric flatfoot proforma

AU Australia

NZ New Zealand

UK United Kingdom

FPI Foot Posture Index

RCSP Resting Calcaneal Stance Position

NCSP Neutral Calcaneal Stance Position

WB Weight Bearing

WHO-ICD World Health Organisation—International Classification of Diseases

MTSS Medial Tibial Stress Syndrome

STJ Subtalar joint

MTJ Midtarsal joint

UCBL University of California Biomechanics Laboratory

RCTs Randomised Controlled Trials

Additional Information and Declarations

Competing Interests

Author Contributions

Human Ethics

Data Availability

The authors declare there are no competing interests.

Sindhrani Dars conceived and designed the experiments, performed the experiments, analyzed the data, contributed reagents/materials/analysis tools, prepared figures and/or tables, authored or reviewed drafts of the paper, approved the final draft.

Hayley Uden and Saravana Kumar conceived and designed the experiments, performed the experiments, authored or reviewed drafts of the paper, approved the final draft.

Helen A. Banwell conceived and designed the experiments, performed the experiments, analyzed the data, contributed reagents/materials/analysis tools, authored or reviewed drafts of the paper, approved the final draft.

The following information was supplied relating to ethical approvals (i.e., approving body and any reference numbers):

The study was approved by the University of South Australia’s Human Research Ethics Committee (Protocol no. 0000035501). All participants provided written informed consent.

The following information was supplied regarding data availability:

The raw data are provided as a Supplemental File.

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
