# Peer review of "When, why and how foot orthoses (FOs) should be prescribed for children with flexible pes planus: a Delphi survey of podiatrists"

_PeerJ, doi:10.7717/peerj.4667_

## Round 0.1 · original submission · Major Revisions

Dear authors

I received comments on your paper submitted to PeerJ. I invite you to address each of the points raised by the reviewers and resubmit your paper as a revised version.

Reviewer 1 ·

Basic reporting

please see below

Experimental design

please see below

Validity of the findings

please see below

Additional comments

Thanks for the opportunity to review this very interesting and comprehensive paper. There is certainly a need to provide consensus and agreement on a the different clinical interventions currently offered for a flexible pes planus in children.
Congratulation on the author and the team for completing such a extensive Delphi survey.
Please find below some minor comments for your possible consideration:

Line 31-32. Would it be possible to actually provide a percentage on how frequently the different type of corrections are prescribed. For example: vertical [heel] cast pour (XX%); minimal arch fill (XX%), forefoot balanced to perpendicular (XX%). I wonder if you were able to retried this data. If so, it might be helpful for the reader.

Line 36 - I would suggest to remove “First of its kind on this topic” --- possibly argue this within the discussion section, but not on the conclusion part of the abstract.
Very comprehensive background Lit Review.it provides a very strong introduction to your paper. In line 77, “Stretch exercises” - could you possible be more specific on what soft tissue were considered for stretches as part of ‘non surgical treatment’ for pes planus.

Line 108: were you able to acquire data that the pods were ‘still currently practising’. If so, please add it within the sentence. A registered podiatrist, may not necessarily means that he/she is still regularly seeing patients

Line 114: could you please add the % of participants recruited from Australia (%), NZ (%), and UK

Line 119: regarding ‘Academic institutes”. Given that academics that teach within the same institution may needs to provide a standard approach of teaching / curriculum to all their students; therefore, they may be inclined to use similar clinical approaches. It could be argued that, for example: if you recruited 5 ‘experts’ from the same university in the UK, they may all provide you with very similar answers, especially on the type of devices that they have access to / or that their department that they can afford. Could you possibly specify if more than one participant actually worked within the same institutions. Obviously for confidentiality reasons there is no need to specify what institution. This additional information, may be useful for the readers, so they can be aware on how heterogenous and varied the participants were as part of your data collection.
Line 124: “informed consent” - could you please provide a short sentence with more information on how the consent was obtained. Was it electronically / separate form / signed / implied - please clarify.

Line 139: could you please indicate from which country the two podiatrists were working on ? and if they were part of your existing team. This will certainly promote more clarity on how the preliminary survey were obtained.

Line 154: could you possibly specify if participants were asked to declare any possible conflict of interest ? This could possibly be perceived as they were potentially trying to ‘promote’ certain type of FOs as part of the data collection stage. If you were not able to capture this info, please add a short statement as part of the limitation section.

Line 173: please add a REF justifying this methodological approach of excluding the statements that did not reach 50%.

Line 200: congratulation for gathering the gender data. However, I would suggest that you should not words it as bias. Please remove the "there was a slight gender bias towards”. Just report : “ overall, male participant …..”

Line 381: regarding Pre-Fabricated FOs - could you also mentioned if the participants reported about the facts that they can be supplied at ‘chair-side’ on the same day as the initial biomechanical consultation.

Line 394 - please remove “as with any research”. Good Limitation section, very comprehensive and informative for the reader. well done

Hope this may help
Kind regards

·

Basic reporting

The purpose of study was to garner consensus and agreement, with Delphi consensus survey, from experienced podiatrists of Australia (AU), New Zealand (NZ) andUnited Kingdom (UK), on the presentation and management of paediatric flexible pes planus.

The secondary aim was to develop a clinical protocol based on this expert opinion to direct clinicians on when, why and how FOs should be prescribed for children with symptomatic 96 flexible pes planus.

The manuscript is very clear, good structure and professional language are used. The references support the arguments and provide sufficient background.





.

Experimental design

A Delphi survey consisting of three rounds was conducted where fifteen podiatrists participants’ opinion was sought. Australia, New Zealand and the United Kingdom were chosen due to similarities in podiatry undergraduate education, scope of podiatric practice and health care contexts. Data was collected using online survey platform

The outcomes of interest were consensus and agreement. All included consensus and agrément statements were compiled to construct ‘A clinical protocol for paediatric flexible pes planus’, that consists of three main sections: Confirm diagnosis; Signs and symptoms; and Intervention.

Research question well defined and relevant . This added value compared to previous similar research that focused on adults. Good description of methods and results.

Validity of the findings

The authors suggest his clinical protocol can be used concurrently with alternative tools such as the pFFP by Evans (2008) and Harris et al. 2004 flow chart, to help clinicians follow a logical pattern for management of paediatric flexible pes planus.

Some limitations are identified by authors to this study like the absence of
agreement on a universally accepted definition of paediatric flexible pes planus and some methodological limitations of conducting a Delphi survey method include risks of collusion affecting participants’ anonymity and absence of criteria for defining an ‘expert’ in the field.

Results are robust and conclusion are well stated, linked to original study questions.

Additional comments

The authors have really well addressed the research questions. Overall, well-done, This added value compared to previous research.

---

## Round 0.2 · accepted · Accept

Dear authors

The reviewers evaluated again your paper and no further comments were provided that might require a revision. Therefore, I would like to congratulating you for having your paper accepted for publication.

Reviewer 1 ·

Basic reporting

please see below

Experimental design

please see below

Validity of the findings

please see below

Additional comments

thanks for kindly including all the suggested changes so quickly.
I believe the paper now looks even stronger
wish you all the best with your future research
kind regards

·

Basic reporting

No comments

Experimental design

No comments

Validity of the findings

No comments

Additional comments

No comments